# Low-dose interleukin-2 in patients with stable ischaemic heart disease and acute coronary syndromes (LILACS): protocol and study rationale for a randomised, double-blind, placebo-controlled, phase I/II clinical trial

Tian Xiao Zhao,[1,2] Michalis Kostapanos,[2] Charmaine Griffiths,[3] Emma L Arbon,[3] Annette Hubsch,[2] Fotini Kaloyirou,[2] Joanna Helmy,[2] Stephen P Hoole,[4] James H F Rudd,[1] Graham Wood,[5] Keith Burling,[6] Simon Bond,[3] Joseph Cheriyan,[2,3] Ziad Mallat[1]

► Prepublication history and additional materials for this paper are available online. To view these files, please visit the journal online (http://dx.doi.org/10.1136/bmjopen-2018-022452).

For numbered affiliations see end of article.

**Correspondence to**
Dr Tian Xiao Zhao;
txz20@cam.ac.uk

## ABSTRACT

**Introduction** Inflammation and dysregulated immune responses play a crucial role in atherosclerosis, underlying ischaemic heart disease (IHD) and acute coronary syndromes (ACSs). Immune responses are also major determinants of the postischaemic injury in myocardial infarction. Regulatory T cells ($CD4^+CD25^+FOXP3^+$; Treg) induce immune tolerance and preserve immune homeostasis. Recent in vivo studies suggested that low-dose interleukin-2 (IL-2) can increase Treg cell numbers. Aldesleukin is a human recombinant form of IL-2 that has been used therapeutically in several autoimmune diseases. However, its safety and efficacy is unknown in the setting of coronary artery disease.

**Method and analysis** Low-dose interleukin-2 in patients with stable ischaemic heart disease and acute coronary syndromes is a single-centre, first-in-class, dose-escalation, two-part clinical trial. Patients with stable IHD (part A) and ACS (part B) will be randomised to receive either IL-2 (aldesleukin; dose range $0.3$–$3 \times 10^6$ IU) or placebo once daily, given subcutaneously, for five consecutive days. Part A will have five dose levels with five patients in each group. Group 1 will receive a dose of $0.3 \times 10^6$ IU, while the dose for the remaining four groups will be determined on completion of the preceding group. Part B will have four dose levels with eight patients in each group. The dose of the first group will be based on part A. Doses for each of the subsequent three groups will similarly be determined after completion of the previous group. The primary endpoint is safety and tolerability of aldesleukin and to determine the dose that increases mean circulating Treg levels by at least 75%.

**Ethics and dissemination** The study received a favourable opinion by the Greater Manchester Central Research Ethics Committee, UK (17/NW/0012). The results of this study will be reported through peer-reviewed journals, conference presentations and an internal organisational report.

**Trial registration number** NCT03113773; Pre-results.

## Strengths and limitations of this study

► The double-blind, placebo-controlled design will allow assessment of the safety and efficacy of low-dose interleukin (IL)-2 in patients with stable ischaemic heart disease and in acute coronary syndrome patients, both conditions where it is currently contraindicated.

► The adaptive dose design of this study will allow assessment of the potential for low-dose IL-2 to increase mean circulating Treg levels by at least 75%.

► Due to its early phase design, this study is not powered to assess any clinical outcome data for patients.

## BACKGROUND

Despite major advances in the treatment, ischaemic heart disease (IHD) remains a significant cause of mortality and morbidity. It is now firmly established that inflammation and the immune response are crucial to the pathophysiology of IHD. This is true both in atherosclerosis, which underlies stable angina, and in progression to plaque instability and disruption in acute coronary syndrome (ACS).[1]

Although the innate immune system has been better studied in atherosclerosis, the role of the adaptive immune responses is now being increasingly understood. Several studies reported a perturbation of the T cell repertoire in ACS patients[2] with expansion of an effector and activated T cell subset[3] which is, at least in part, directed to antigens contained in the disrupted plaque.[4] Initial preclinical findings have shown that

regulatory T (Treg) cell-mediated immunity reduces experimental atherosclerosis and plaque inflammation.[5]

Even though Treg cells in humans are less distinct and more heterogeneous than in mice,[6] there is evidence to demonstrate their role in IHD. In patients with ACS, there is an imbalance between T effector and Treg cells. Despite the effector T cell compartment activation, the percentage and function of circulating Tregs appear to be significantly decreased in in the setting of ACS.[7–10] This imbalance is thought to play a key role in coronary plaque progression and destabilisation. In this context, low levels of circulating baseline $CD4^+Foxp3^+$ Treg cells were associated with an increased risk for acute coronary events in the Malmö Diet and Cancer Study.[11]

Following myocardial infarction, the ischaemic and necrotic myocardial tissue may present self-antigens to the immune system, leading to antigen-specific, autoimmune adaptive responses.[12 13] Recent studies indicate that $CD4^+$ T cells, and particularly Treg cells, are important for the control of postischaemic immune responses and the promotion of myocardial healing.[13–16] In another study, inhibition of Treg recruitment to the site of myocardial injury resulted in excessive postischaemic inflammation, matrix degradation and adverse remodelling.[14] In contrast, in vivo expansion of Treg cells or their therapeutic activation by superagonistic anti-CD28 antibodies attenuated left ventricular remodelling and improved cardiac function.[15 16]

Interleukin-2 (IL-2) plays a key role in Treg cell development, expansion, survival and suppressive function.[17 18] Deficiency of either IL-2 or IL-2 receptor in mice greatly compromises Treg development and promotes autoimmune responses.[19] Supplementation of IL-2 substantially increases Treg cell levels and significantly limits plaque development and inflammation in mice prone to atherosclerosis.[20 21] Treg cells show a much lower threshold response to IL-2 receptor signalling compared with effector T cells. This led to the hypothesis that, in contrast to high-dose IL-2 designed to activate T effector cells in cancer, supplementation with low doses of IL-2 in the setting of T cell-mediated immune diseases could selectively promote the expansion of Treg cells at the expense of T effector cells, thereby limiting harmful immune responses. This hypothesis was initially confirmed in two pilot human clinical studies in two different disease settings: graft-versus-host disease[22 23] and in hepatitis C virus-induced vasculitis.[24] In both studies, administration of low doses of IL-2 in the form of aldesleukin (daily administration of $0.3×10^6$ to $3.0×10^6$ IU IL-2 per square metre of body surface area for 8 weeks, or repetitive 5-day courses of $1.0×10^6$ to $3.0×10^6$ IU IL-2) led to a rapid and marked expansion of the circulating pool of Treg cells, which were at least doubled without affecting the pool of conventional $CD4^+$ T (ie, T effector) cells. The expanded Tregs retained potent suppressive functions, and the treatment was associated with a reduction in the inflammatory response and a concomitant clinical improvement in a substantial proportion of patients. Treatment

with low-dose IL-2 was safe, and no adverse effects were reported. This strategy is currently being adapted and tested in various disease settings, where Treg cell promotion is believed to be of potential therapeutic benefit.[23–26] In this trial, we hypothesise that low-dose IL-2 (aldesleukin) can be used in IHD to increase Treg numbers and to rebalance the immune system with the overall goal of decreasing recurrent myocardial infarction and cardiovascular death.

Aldesleukin (Proleukin, Novartis) is a commercially available IL-2 licenced for the treatment of metastatic renal cell carcinoma in the UK. It is produced by recombinant DNA technology using an *Escherichia coli* strain, which contains a genetically engineered modification of the human IL-2 gene, and is administered either intravenously or subcutaneously (SC). Following short intervenous infusion, its pharmacokinetic profile is typified by high plasma concentrations, rapid distribution into the extravascular space and a rapid renal clearance. The recommended doses for continuous infusion and subcutaneous injection (as detailed in the Summary of Product Characteristics) are repeated cycles of $18×10^6$ IU per $m^2$ per 24 hours for 5 days and repeated doses of $18×10^6$ IU, respectively. Peak plasma levels are reached in 2–6 hours after SC administration, with bioavailability of aldesleukin ranging between 31% and 47%. The process of absorption and elimination of subcutaneous aldesleukin is described by a one-compartment model, with a 45 min absorption half-life and an elimination half-life of 3–5 hours.[27]

## Use of IL-2 in clinical trials to date

The first report of effective IL-2 therapy in human cancer trials was published in 1985.[28] The trial patients in that study were placed on dose-escalated IL-2 regimens, of up to approximately 120 million IU (MIU). Associated with these high IL-2 doses were side effects such as capillary leak syndrome (which is characterised by a loss of vascular tone and extravasation of plasma proteins and fluid into the extravascular space, ultimately resulting in hypotension, tachycardia, dyspnoea and pulmonary oedema) and kidney and liver damage (both characterised by increased serum creatinine and bilirubin levels, respectively)[29].

The use of low dose IL-2 to expand Treg cell populations in autoimmune and alloinflammatory conditions has been previously explored and published in human clinical trials. In these studies, patients received at least 1 dose of IL-2 ranging from $0.3×10^6$ IU to $3.0×10^6$ IU. In two studies of 12 and 21 healthy volunteers, respectively, there were minimal adverse events (AEs), consisting mainly of grade 1 injection site reactions. No cardiovascular AEs were noted.[30 31] In one phase I/IIa study, 24 patients with diabetes mellitus were recruited and given a maximum dose of $3.0×10^6$ IU daily for 5 days. The authors found that IL-2 was well tolerated at all doses, with no serious adverse events (SAEs). However, there was a dose–response relationship for non-serious AEs. The most common AEs in the treatment phase were injection-site reactions and an influenza-like syndrome.[32] In a later trial

of 40 indivisuals with type 1 diabetes, the authors found that doses of aldesleukin were well tolerated at all doses, with no SAEs reported. The majority of participants had an expected AE at the injection site consisting a non-itchy, local (1–5 cm), non-painful erythematous rash that resolved on average by day 10.[33] No cardiovascular AEs were reported in either study. Low dose IL-2 has also been used in 38 patients with systemic lupus erythematosus (SLE)[26] who are considered to have a higher risk of coronary artery disease and therefore cardiac events.[34] However, no SAEs were observed while injection site reactions and influenza-like symptoms were observed in 13.2% and 5.3% of patient, respectively.[26]

Nevertheless, IL-2 is contraindicated in patients with a significant history, or current evidence of, severe cardiac disease. Therefore, we sought to determine the safety and efficacy of low-dose aldesleukin in patients with pre-existing cardiac conditions. A detailed and conservative risk mitigation strategy was adopted to ensure patient safety was maintained throughout the conduct of the trial (see online supplementary file 1). We hypothesise that low-dose IL-2, unlike higher doses, can be safely administered and is effective in expanding the Treg population in patients with stable and acute coronary artery disease. In this trial, low-dose IL-2 will initially be administered in patients with stable IHD at escalating doses and, following safety reviews, will be given to patients with ACS. The Treg response data from the ACS population will help select the most appropriate dose to assess efficacy in future clinical trials in ACS.

## METHOD

This is a an academically driven, prospective single-centre, randomised, double-blind, placebo-controlled, phase I/II clinical trial. It will be performed at the National Institute for Health Research/Wellcome Trust Cambridge Clinical Research Centre, Cambridge University Hospitals, Cambridge, UK, with overall study coordination provided by the Cardiovascular Trials Office of the Cambridge Clinical Trials Unit, Cambridge University Hospitals National Health Service (NHS) Foundation Trust. The study is sponsored by Cambridge University Hospitals NHS Foundation Trust.

## Study populations

The trial will be performed in two parts. Part A will include patients with stable IHD aged 18–75 years with a clinical diagnosis of IHD for more than 6 months (ascertained by either having had a previous diagnosis of MI or having symptoms of angina and a coronary angiogram showing obstructive (stenosis >50%) coronary disease). The inclusion and exclusion criteria are detailed in table 1 but, in summary, patients with a myocardial infarction within the last 6 months, cardiogenic shock, hypotension/hypertension, heart failure (ejection fraction (EF) <45%), proarrhythmogenic conditions, renal, hepatic, thyroid or haematological dysfunction, active infection, poorly controlled diabetes, active autoimmune disease, current malignancy, history of seizures or immunosuppression will be excluded from this part of the study.

Part B will be performed after part A and will include patients aged of 18–85 years admitted with a diagnosis of non-ST elevation myocardial infarction. The inclusion and exclusion criteria are detailed in table 1. In brief, patients with cardiogenic shock, hypertension/hypotension, heart failure (EF <35%), long QT or arrhythmias, renal, hepatic, thyroid or haematological dysfunction, active infection, active autoimmune disease, current malignancy, history of seizures or immunosuppression will be excluded from this part of the study.

## Study protocol
### Part A

Patients will be recruited from advertisements, outpatient clinics or research databases. Participants will have at least 24 hours to review the patient information sheet prior to informed consent. Study procedures will only be conducted following formal written consent at the screening visit 1 (V1). Baseline blood tests will consist of electrolytes, renal, liver and thyroid function, full blood count, clotting profile, hepatitis B/C and HIV screening, HbA1c and pregnancy screen (where applicable). Baseline vitals, ECG, echocardiogram, clinical history and physical examination will also be performed. Randomisation will be carried out via a paper-based concealment list generated by a statistician. To maintain the overall quality and legitimacy of the clinical trial, unblinding will only occur in exceptional circumstances when knowledge of the actual treatment is essential for further clinical management of the patient.

The trial design is described in detail in figure 1. In brief, following randomisation at V1, patients will attend five consecutive daily outpatient visits (V2–6) during which blinded subcutaneous injections of aldesleukin or placebo will be administered. At each visit, prior to the study drug administration, the medical history will be obtained/reviewed, along with a physical examination, baseline vitals, safety bloods and a 12-lead ECG. Patients will have continuous cardiac monitoring during each visit for at least 30 min predose and 1.5 hours postdose. After dosing, a series of ECGs will be performed at 15 min, 30 min and 60 min, while vitals will be assessed at 30 min and 60 min. For each dosing visit, serum IL-2 levels will be taken at baseline and at 90 min postdose.

There are two follow-up visits (V7 and V8). Assessments during both visits will include a medical review, physical examination, vitals, ECGs and safety bloods tests. Additionally, at V8, a follow-up thyroid function blood test and echocardiogram will be performed.

In addition, during visits V2 and V7 (figure 1), cardiac biomarkers (high sensitivity C reactive protein (hs-CRP), IL-6, brain-type natriuretic peptide (BNP) and troponin) and lymphocyte subsets (including Treg level, see below) analysis will be performed. During visit V8, cardiac biomarkers will be reaassessed.

**Table 1** Trial inclusion and exclusion criteria for parts A and B

| Part A | Inclusion criteria | ▲ Age 18–75 years old inclusive. |
| --- | --- | --- |

Inclusion criteria:
- ▲ Age 18–75 years old inclusive.
- ▲ Previous history (>6 months from planned first day of dosing) of coronary artery disease.
- ▲ No history of recent (<6 months) admissions for an unstable cardiovascular event, for example, MI, unstable angina, ACS.
- ▲ Written informed consent for participation in the trial.

Exclusion criteria:
- ▲ Current presentation with cardiogenic shock (systolic blood pressure <80 mm Hg, unresponsive to fluids or necessitating catecholamines), severe congestive heart failure and/or pulmonary oedema.
- ▲ Known active bleeding or bleeding diatheses.
- ▲ Known active infection requiring antibiotic treatment.
- ▲ Severe haematological abnormalities (haematocrit <30% and platelet cell count of <100 × 10³/µL and white cell count <4 × 10³/µL).
- ▲ Known malignancies requiring active treatment or follow-up (however, patients with current/a history of localised basal or squamous cell skin cancer are not excluded from participation in this trial).
- ▲ Known heart failure with impaired LV function (LV EF<45%).
- ▲ Hypotension (systolic BP <100 mm Hg, DBP <50 mm Hg) at screening.
- ▲ Uncontrolled hypertension (>160/100 mm Hg) at screening.
- ▲ History of recurrent syncope (electrocardiographic history suggestive of arrhythmia syncope (eg, bifascicular block, sinus bradycardia <40 beats per minute in absence of sinoatrial block or medications, pre-excited QRS complex, abnormal QT interval, ST segment elevation leads V1 through V3 (Brugada syndrome), negative T wave in right precordial leads and epsilon wave (arrhythmogenic right ventricular dysplasia/cardiomyopathy)).
- ▲ Known hepatic failure or abnormal LFTs at baseline (ALT >2× ULN).
- ▲ Elevated total bilirubin levels, (TBL >1.5× ULN) and alkaline phosphatase, ALP (ALP >1.5× ULN), at baseline.
- ▲ Acute kidney injury or chronic kidney disease at stage >3B (eGFR <45 mL/min/1.73 m²).
- ▲ Respiratory failure.
- ▲ Known hyperthyroidism or hypothyroidism.
- ▲ History of drug-induced Stevens Johnson syndrome, drug reaction with eosinophilia and systemic symptoms (DRESS syndrome) or toxic epidermal necrolysis.
- ▲ History of recurrent epileptic seizures in the previous 4 years; repetitive or difficult to control seizures, coma or toxic psychosis lasting >48 hours.
- ▲ If known diabetic, uncontrolled diabetes defined as HbA1c >64 mmol/mol.
- ▲ Average corrected QT interval >450 ms using Bazett's formula using triplicate ECGs (or >480 ms if bundle branch block).
- ▲ Known chronic active hepatitis (B or C).
- ▲ Known HIV infection.
- ▲ Current infection possibly related to recent or ongoing immunosuppressive treatment.
- ▲ Known autoimmune disease requiring active immunosuppressive therapy.
- ▲ History of organ transplantation.
- ▲ Any oral or intravenous immunosuppressive treatment including prednisolone, hydrocortisone or disease-modifying drugs such as azathioprine, interferon-alpha, cyclophosphamide or mycophenolate. (Other immunosuppressive therapies should be discussed with PI. Inhaled or topical steroids are permissible).
- ▲ Known pregnancy at screening or visit 2 (where applicable).
- ▲ Ongoing lactation.
- ▲ Inability to comply with trial procedures.
- ▲ Current participation in other interventional clinical trials.
- ▲ Contra indication to interleukin (IL)-2 treatment or hypersensitivity to IL-2 or to any of its excipients.
- ▲ Unwillingness or inability to provide written informed consent for participation.
- ▲ Any medical history or clinically relevant abnormality that is deemed by the principal investigator/delegate and/or medical monitor to make the patient ineligible for inclusion because of a safety concern.

Continued

**Table 1** Continued

| Part B | | |
|---|---|---|
| | Inclusion criteria | ▲ Age 18–85 years old inclusive. |
| | | ▲ Current admission (on at least screening visit) with acute coronary syndrome (non-ST elevation myocardial infarction, ie, NSTEMI or unstable angina) with symptoms of myocardial ischaemia lasting 10 min or more with the patient at rest or with minimal effort plus either elevated levels of TnI on admission or dynamic changes in ECG (new ST-T changes) or T-wave inversion. |
| | | ▲ Willingness to be dosed within 8 days from initial date of current admission for ACS. |
| | | ▲ Written informed consent for participation in the trial. |
| | Exclusion criteria | ▲ ST elevation myocardial infarction (heart attack) on this admission. |
| | | ▲ Current presentation with cardiogenic shock (systolic blood pressure <80 mm Hg, unresponsive to fluids, or necessitating catecholamines), electrical instability, severe congestive heart failure and/or pulmonary oedema. |
| | | ▲ Known active bleeding or bleeding diatheses. |
| | | ▲ Known active infection requiring antibiotic treatment. |
| | | ▲ Severe haematological abnormalities (haematocrit <30% and platelet cell count of <100 × 10³/µL and white cell count <4 × 10³/µL). |
| | | ▲ Known malignancies requiring active treatment or follow-up (however, patients with current/a history of localised basal or squamous cell skin cancer are not excluded from participation in this trial). |
| | | ▲ Known heart failure with impaired LV function with LV EF <35%. |
| | | ▲ Hypotension (systolic BP <100 mm Hg, DBP <50 mm Hg). |
| | | ▲ Uncontrolled hypertension (>160/100 mm Hg) at screening. |
| | | ▲ History of recurrent syncope (electrocardiographic history suggestive of arrhythmia syncope (eg, bifascicular block, sinus bradycardia <40 beats per minute in absence of sinoatrial block or medications, pre-excited QRS complex, abnormal QT interval, ST segment elevation leads V1 through V3 (Brugada syndrome), negative T wave in right precordial leads and epsilon wave (arrhythmogenic right ventricular dysplasia/cardiomyopathy)). |
| | | ▲ Known hepatic failure or abnormal LFTs at baseline (ALT >2× ULN). |
| | | ▲ Elevated TBL (TBL >1.5× ULN) and ALP (ALP >1.5× ULN), at baseline. |
| | | ▲ Renal impairment at screening (creatinine clearance (Cockcroft-Gault) <45 mL/min). |
| | | ▲ Acute respiratory failure. |
| | | ▲ Known hyperthyroidism or hypothyroidism. |
| | | ▲ History of drug-induced Stevens Johnson syndrome, DRESS syndrome or toxic epidermal necrolysis or contrast allergy (requiring steroid treatment). |
| | | ▲ History of recurrent epileptic seizures in the previous 4 years, repetitive or difficult to control seizures, coma or toxic psychosis lasting >48 hours. |
| | | ▲ Average corrected QT interval >450 ms using Bazett's formula using triplicate ECGs (or >480 ms if bundle branch block). |
| | | ▲ Known chronic active hepatitis (B or C). |
| | | ▲ Known HIV infection. |
| | | ▲ Current infection possibly related to recent or ongoing immunosuppressive treatment. |
| | | ▲ Known autoimmune disease requiring active immunosuppressive therapy. |
| | | ▲ History of organ transplantation. |
| | | ▲ Any oral or intravenous immunosuppressive treatment including prednisolone, hydrocortisone or disease-modifying drugs such as azathioprine, interferon-alpha, cyclophosphamide or mycophenolate. (Other immunosuppressive therapies should be discussed with PI. Inhaled or topical steroids are permissible.) |
| | | ▲ Known pregnancy at screening. |
| | | ▲ Ongoing lactation. |
| | | ▲ Inability to comply with trial procedures. |
| | | ▲ Current participation in the active dosing phase of other interventional clinical trials. |
| | | ▲ Contra indication or hypersensitivity to IL-2 treatment or to any of its excipients. |
| | | ▲ Unwillingness or inability to provide written informed consent for participation. |
| | | ▲ Any medical history or clinically relevant abnormality that is deemed by the principal investigator/delegate and/or medical monitor to make the patient ineligible for inclusion because of a safety concern. |

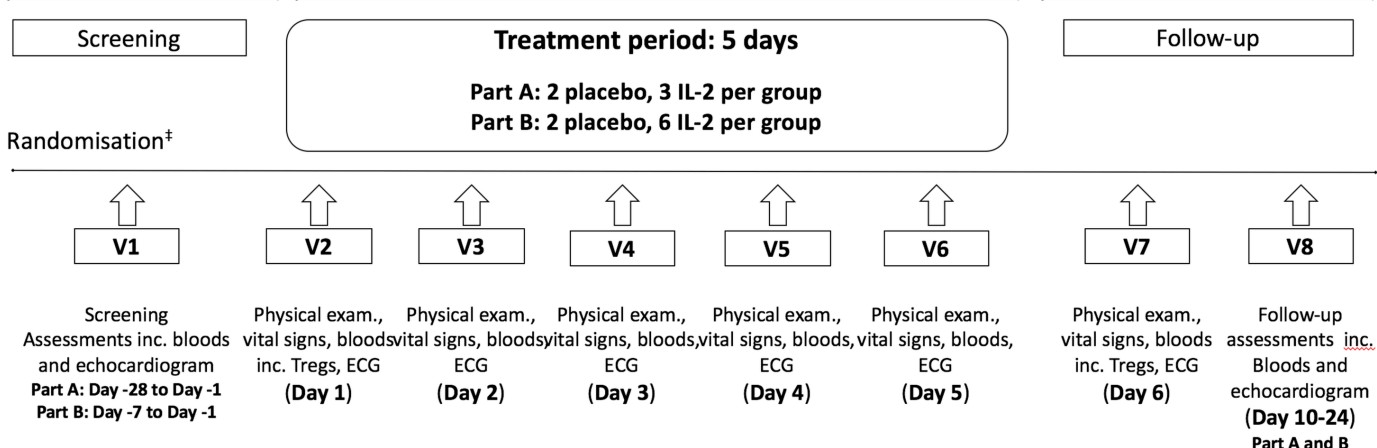

**Figure 1** Trial design per patient. Each patient will make a total of eight study visits. IL-2, interleukin; V, visit.

A total of 25 patients, five in each of the five dosing groups, will be included in part A (drug:placebo ratio of 3:2). In line with current phase I trial designs, a sentinel dosing approach will be employed whereby the first two patients of each group will be allocated to either aldesleukin or placebo in a random order. After a blinded safety review of these first two patients, the remaining patients will then be dosed (see figure 2).

### Part B

Patients admitted with a primary diagnosis of non-ST elevation myocardial infarction will be recruited from the medical and cardiology wards at Cambridge University Hospitals NHS Foundation Trust. Patients may continue to receive trial treatments if they are transferred to the local interventional centre at Royal Papworth Hospital. Participants will be given at least 24 hours to review the patient information sheet prior to formal consent.

Dosing should commence within 8 days of screening. All visits and blinding procedures will be the same as part A. However, in part B, a total of 32 patients will be recruited, eight patients in each of the four dosing groups (drug:placebo ratio of 6:2). A sentinel approach to dosing will also be employed in each group. After a blinded safety review of the first two patients, the remaining cohort will be randomly allocated to study treatments as shown in figure 3. The visit schedule for each patient is the same as part A (figure 1).

### Dose-escalation strategy

The first group of patients in part A will receive $0.3 \times 10^6$ IU of aldesleukin daily. Thereafter, a blinded review of patient data by the blinded Trial Management Group (TMG) including review of AEs, blood results, ECGs, clinical records and, where possible, drug pharmacodynamics and pharmacokinetics. The TMG will be composed of

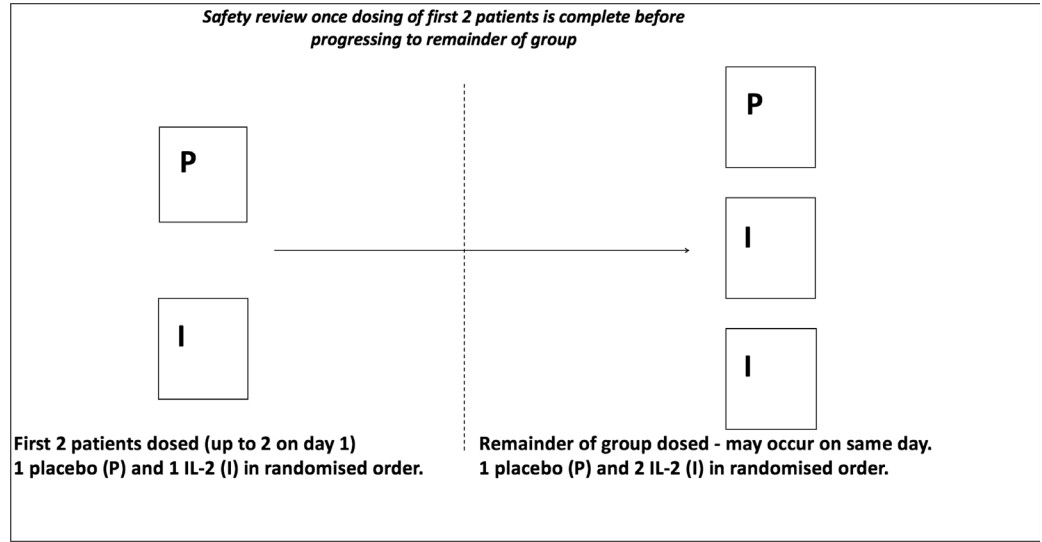

**Figure 2** Trial design for each group in part A. There are a total of five dose levels in part A.

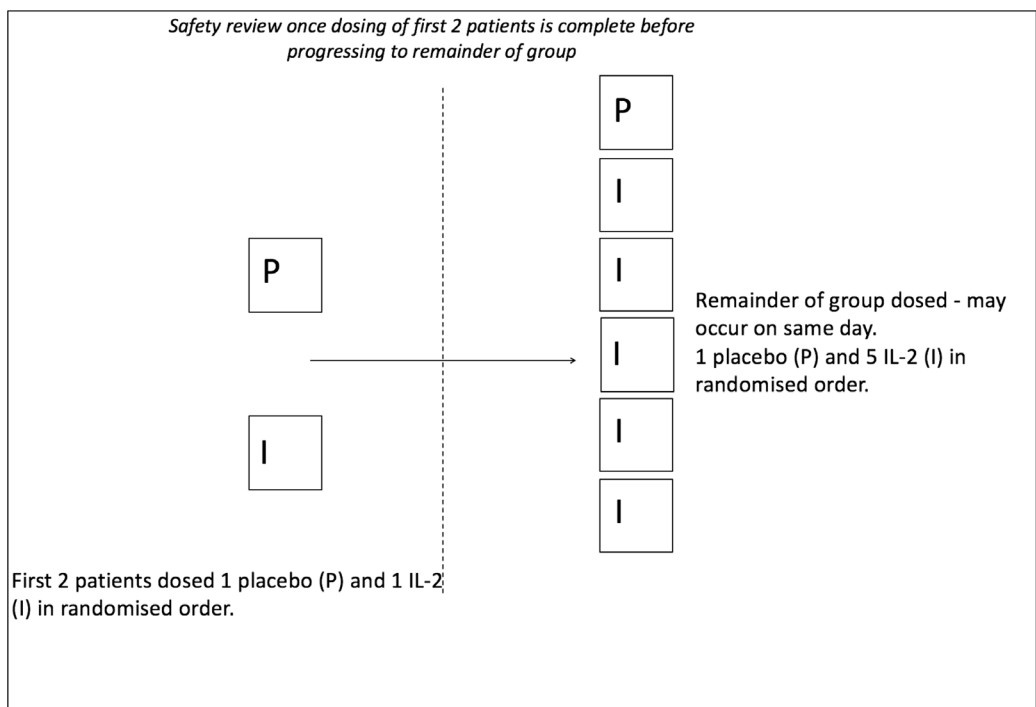

*Safety review once dosing of first 2 patients is complete before progressing to remainder of group*

P

P

I

I

Remainder of group dosed - may occur on same day.
1 placebo (P) and 5 IL-2 (I) in randomised order.

I

I

I

I

First 2 patients dosed 1 placebo (P) and 1 IL-2 (I) in randomised order.

**Figure 3** Trial design for each group in part B. There are a total of four dose levels in part B.

an experienced, accredited early phase lead physician (LILACS chief investigator), a research physician/scientist, a research nurse, trial coordinator and an unblinded study statistician (for the purposes of data analysis). All data presented by the unblinded statistician to the TMG will be in an aggregated format to preserve the blind for the other TMG members. This is consistent with the commercial standard for such early phase trials in industry. The dose in the second group will be determined after this review, and the same process will be followed in each of the following groups. The maximum dose increments allowed by the protocol between groups will be double the previous dose and capped to a maximum $3.0 \times 10^6$ IU.

A robust set of specific and general withdrawal criteria, as well as objective stopping criteria have been put into place to maintain the safety:risk benefit, in particular due to the risk of capillary leak syndrome.

Following completion of part A, an unblinded independent Data Monitoring Committee (DMC) will review all available safety data, together with any other analyses that the committee may request. The DMC will be composed of an independent group of clinical researchers with suitable experience in experimental medicine and early phase clinical trials. These researchers are independent from the trial team and have not been involved in the setup or running of this clinical trial. They will determine whether it is safe to progress to part B, based on available safety and pharmacodynamic data provided by the unblinded statistician. After this analysis, the dose in each group will be determined based on the review of ongoing patient data by the TMG, as in part A previously. The protocol mandates that the maximum dose used in part B will not exceed that of part A. DMC will be governed

by a charter set up a priori and signed up by all members prior to the commencement of the trial (see supplementary file 2).

## Outcome measure
### Part A
The primary outcome will be the safety of IL-2 in patients. This will be assessed through:
► A review of AEs and SAEs, and concomitant medications.
► Changes in safety bloods (electrolytes (sodium, potassium and urea), bone profile (calcium and phosphate), serum creatinine, liver function tests (alanine transaminase, aspartate transaminase, alkaline phosphatase, bilirubin and gamma glutamyl transferase (GT)), thyroid function tests (thyroid stimulating hormone), blood glucose, full blood count and differential, and clotting (prothrombin time and activated partial thromboplastin time).
► Twelve-lead ECG and cardiac monitoring changes (arrhythmias, ischaemic changes and QTcB).
► Vital observations (blood pressure, heart rate, respiratory rate, peripheral oxygen saturation and temperature).
► Echocardiogram changes at baseline and follow-up.
  Exploratory endpoints will include:
► Change in the mean circulating Treg level measured by fluorescence activated cell sorting (FACS) analysis following treatment with IL-2, over the 5 days of the treatment period.
► Change in cardiac biomarker measurements including hs-CRP, troponin I, IL-6 and b-type natriuretic peptide) from analysed blood samples.

► Change in lymphocyte subsets measured by FACS analysis.
► Pharmacokinetic analysis of IL-2 levels.

### Part B

As with part A, the primary endpoint will be safety and tolerability of IL-2. A further primary endpoint will be the change in mean circulating Treg levels and whether IL-2 increases mean circulating Treg levels by at least 75% over the 5 days of the treatment period. Exploratory endpoints are the same as for Part A.

### AE reporting

AEs, adverse reactions (ARs), serious AE/ARs (SAEs/SARs) and suspected unexpected SARs will be defined as per the International Conference on Harmonisation definitions. A suitably qualified medical doctor will determine the relationship and causality of each AE to the study drug as either 'related' (defined as having a plausible temporal relation and not judged attributable to other causes) or 'not related'. They will also make an assessment on severity and seriousness. Abnormal or significant changes in laboratory results are only AEs if they are deemed to be of clinical significance or if a medical intervention is required. Whether a patient needs to be withdrawn due to the severity of their AE is left up to the discretion of principal investigator.

### Lymphocyte analysis

Lymphocyte subset analysis will be performed at the Department of Clinical Immunology, Cambridge University Hospitals, Cambridge, UK, within 4 hours of sample collection in EDTA. Laboratory technicians will be blinded to treatment allocation. The antibodies that will be used are anti-CD3 (clone SK7, phycoerythrin (PE)-Cy7 labelled; BD Biosciences), anti-CD4 (clone RPA-T4, fluorescein Isothiocyanate (FITC) labelled; BD Biosciences), anti-CD127 (clone HIL-7R-M21, PE labelled; BD Biosciences), anti-CD25 (clone M-A251 and 2A3, allophycocyanin (APC) labelled; BD Biosciences), anti-CD45RA (clone HI100, APC-Cy7 labelled; BioLegend) and anti-CD62L (clone DREG-56, PerCP/Cy5.5 labelled; BioLegend). Whole blood will be assessed by performing clinical FACS to measure the absolute lymphocyte count, lymphocytes subsets and CD25 expression on Treg cells. Fixed concentrations of fluorescently labelled beads will be added to whole blood to count the absolute number of lymphocytes, $CD3^+$, $CD4^+$ and $CD8^+$ T cells, $CD19^+$ B cells and $CD19^-$ $CD16^+$ $CD56^+$ NK. Simultaneous whole blood FACS assay will be performed where a lymphocyte gate is drawn to include all events, while the $CD3^+$, $CD4^+$ T-cell gate excludes $CD8^+$ T cells and B cells. Six standardised beads labelled with different quantities of fluorescent APC are measured by FACS to accurately measure CD25-APC on the surface of Tregs compared with a standardised curve. To minimise interassay variation, mean fluorescence intensity can be read from this curve. Tregs will be defined by $CD3^+CD4^+CD25^{high}CD127^{low}$ makers

and will be separated from non-Tregs and used to calculate the absolute Treg count out of $CD3^+CD4^+$ T cells. Among the non-Treg T effector $CD43^+CD4^+$ population, we will define effector memory cells by $CD45RA^-CD62L^-$ markers, effector memory CD45 $RA^+$(TEMRA) cells by $CD45RA^+CD62L^-$ markers, naïve effectors cells by $CD45RA^+CD62L^+$ markers and central memory cells by $CD45RA^-CD62L^+$ markers. Total memory effectors are the sum of central memory and effector memory cells.

### Cardiac biomarkers

Blood will be taken in gel serum tubes, and the serum will be banked and analysed at the Core Biochemical Assay Laboratory, Cambridge. Hs-CRP and N-terminal pro brain natriuretic peptide (NT-proBNP) will be measured by immunoassays on the Siemens Dimension EXL autoanalyser. All reagents and calibrators are supplied by Siemens, and assays will be performed according to the manufacturer's instructions.

IL-6 will be measured in duplicate using ultrasensitive electrochemical luminescence immunoassay on the Mesoscale Discovery assay platform and read on the MesoScale Diagnostics Sector Imager 6000. All reagents and calibrators will be supplied by MesoScale Discovery.

### Stopping criteria

Dose-escalation stopping criteria will be met if two patients within a trial group experience any combination of: a SAE defined as possibly, probably or definitely related to the trial drug (ie, it is a SAR), an adverse event that is severe and at least possibly related to the trial drug or any of the objective stopping criteria detailed box 1. The following will then occur:
► Dosing will be immediately discontinued for the patients experiencing the event.
► Dosing will be halted for all other patients currently in the treatment period of the trial (ie, patients receiving treatment in the same group).
► A safety review by the independent DMC will be conducted to determine how to proceed with the trial.
► Any further single instances of the events outlined above for the same group will trigger a further DMC safety review.
► Any patients who have their dosing discontinued will be withdrawn from the trial.

Additionally, specific objective stopping criteria are set out in box 1, which may trigger an unscheduled DMC review prior to any further dose escalations. Specific and general withdrawal criteria are also listed in online supplementary file 1.

### Safety monitoring committees

The TMG and the DMC are independent of each other. The function of the DMC is delineated in the DMC charter (online supplementary file 2). The chief investigator of the trial will report to the DMC on the course of the trial during open sessions of the DMC. All DMC meetings will be held in private without any involvement

---

**Box 1  Objective stopping criteria triggering a Data Monitoring Committee safety review of dose escalation**

► QTcB >500 ms (or >530 ms if baseline QTcB=450–480 ms) OR QTcB change from baseline >60 ms (based on an average of triplicate electrocardiogram (ECGs)).

► Acute pulmonary oedema or congestive heart failure.

► Symptomatic systolic BP <90 mm Hg and/or diastolic blood pressure (BP) <60 mm Hg OR persistent symptomatic systolic BP 80–90 mm Hg for >15 mins OR severe hypertension (as defined by BP >180/120 mm Hg).

► ST-elevation myocardial infarction (STEMI) occurrence

► Atrial fibrillation with rapid ventricular response >150/min, supraventricular tachycardia or bradycardia that requires treatment or is recurrent or persistent.

► Sustained ventricular tachycardia or ventricular fibrillation.

► Any patient who develops doubling of creatinine.

► Systemic hypersensitivity reaction that cannot be attributed to an identifiable cause.

► If a life-threatening infection is confirmed clinically with a positive microbiological test.

► Signs suggestive of hepatic failure including encephalopathy, increasing ascites, signs of coagulopathy, liver pain and/or tenderness on palpation, hypoglycaemia presumed to be secondary to liver failure and active GI bleeding. Withdrawal also if alanine aminotransferase (ALT)>3 ULN.

► Seizure activity, coma, severe lethargy or somnolence.

► Risk of respiratory insufficiency requiring intubation.

---

of the trial team. The unblinded statistician is the only person who reports to the DMC and is part of the TMG. However, any data presented to the TMG are presented in a manner that maintains the blind.

The TMG will assess safety in a blinded manner on an ongoing basis at regular intervals during the course of the trial. Between part A and part B, the DMC will be formally convened to review the unblinded data with the unblinded statistician to determine whether it is reasonable for the trial to progress to the next stage.

### Statistical methods and data handling

This is an exploratory study that is not designed to formally test a hypothesis in a confirmatory fashion. Given that both parts of the trial have clinical safety as primary endpoints, a formal power calculation is not relevant. A sample size of 57 patients is achievable within the proposed time scale, given the size of the targeted patient population at our study site. The frequency of AEs per patient will be summarised for each event based on dose level. Summary statistics of laboratory values by dose and visit will be produced where required. The statistician will use the data from each group to perform a modelling analysis of the effect of aldesleukin based on dose and effect size using a smoothed line plot of the mean and 95% CIs. Generally summary statistics of continuous variables will report mean, median, SD, min and max, although a log-transformed scale may be used where the data are skewed. Binary or categorical variables will be summarised using the p% (x/n) format. The Treg

data and other secondary biomarker endpoints will be summarised with individual patient profiles over time, and summary statistics will be broken down by dose and visit. Formal estimates of the differences between doses will be made at each time point with accompanying 95% CIs and p values.

Subjects will be coded by a numeric code to create an anonymous dataset. All data will be transferred into a case report form, which will be coded onto a MACRO database. All data will be anonymised and stored encrypted on a secured computer to ensure patient confidentiality.

### Patient and public involvement

Heart attacks are distressing and impacts patients' lives dramatically.[35] The aim of this research is to help ameliorate this issue and potentially reduce reoccurrence. Lay members of the ethics committee reviewed this study and made constructive comments, which have been addressed. Patients were not involved in the recruitment to or conduct of the study. All patients provided full informed consent, with at least 24 hours to consider the information and at least two opportunities to discuss the trial in detail with the investigators. The results of the study will be disseminated to all study patients at the end of the trial.

### Study timeline

The trial began on the 15 May 2017. The anticipated final follow-up visit(s) will be in January 2019. Primary analyses are projected to be completed by February 2019.

**Author affiliations**

[1]Department of Medicine, Division of Cardiovascular Medicine, University of Cambridge Medicine, Cambridge, UK

[2]Division of Experimental Medicine and Immunotherapeutics (EMIT), Department of Medicine, University of Cambridge Medicine, Cambridge, Cambridgeshire, UK

[3]Cambridge Clinical Trials Unit, Cambridge University Hospitals, Cambridge, Cambridgeshire, UK

[4]Department of Interventional Cardiology, Royal Papworth Hospital NHS Trust, Cambridge, UK

[5]Department of Immunology, Cambridge University Hospitals, Cambridge, UK

[6]Clinical Biochemistry, Cambridge University Hospitals, Cambridge, UK

**Acknowledgements**  JC acknowledges funding support from the National Institute for Health Research (NIHR) Cambridge Comprehensive Biomedical Research Centre. ZM acknowledges funding support from the British Heart Foundation (BHF) for his BHF Chair. JHFR acknowledges funding support from the HEFCE, the NIHR Cambridge Biomedical Research Centre, the British Heart Foundation, the EPSRC and the Wellcome Trust. We acknowledge support from the Cambridge Clinical Research Centre and the Core Biochemistry Assay Laboratory in the conduct of this trial and its endpoints.

**Contributors**  TXZ, MK, SPH, JC and ZM contributed to the writing of this manuscript. TXZ, MK, CG, ELA, AH, FK, JH, JHFR, GW, KB, SB, JC and ZM contributed to the writing of the protocol.

**Funding**  This work was funded by the Medical Research Council, grant number MR/N028015/1 and the British Heart Foundation Cambridge Centre of Excellence (RCAG/521). Setup and running of the study and decision to publish results is independent of the funder.

**Competing interests**  None declared.

**Patient consent**  Not required.

**Ethics approval**  The study was given a favourable opinion by the Greater Manchester Central Research Ethics Committee, UK (17/NW/0012) and approved

by the UK's Health Research Authority. The MHRA formally granted regulatory acceptance on 28 April 2017. All study procedures will be conducted after formal written consent, in accordance with the Declaration of Helsinki. The trial was registered on Clinicaltrials.gov (NCT03113773) prior to trial commencement, and the results of this study will be published in a peer-reviewed journal after completion.

**Provenance and peer review** Not commissioned; externally peer reviewed.

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
