## [Reviewer comments · BMJ Open]

ARTICLE DETAILS

TITLE (PROVISIONAL)	Low dose interleukin-2 in patients with stable ischaemic heart disease and acute coronary syndromes (LILACS): protocol and study rationale for a randomised, double-blind, placebo controlled, phase I/II clinical trial
AUTHORS	Zhao, Tian; Kostapanos, Michalis; Griffiths, Charmaine; Arbon, Emma; Hubsch, Annette; Kaloyirou, Fotini; Helmy, Joanna; Hoole, Stephen; Rudd, James; Wood, Graham; Burling, Keith; Bond, Simon; Cheriyan, Joseph; Mallat, Ziad

VERSION 1 – REVIEW

REVIEWER	Peter Libby BWH and HMS Boston, MA, USA
REVIEW RETURNED	18-Mar-2018

GENERAL COMMENTS	This manuscript describes the protocol in study design for a highly innovative approach to immunomodulation in patients with stable or acute coronary artery disease. The premise is using low dose interleukin 2 (IL – 2) to raise the regulatory T cell subset in an effort to combat over exuberant adaptive immune responses in atherosclerosis and particularly in acute myocardial infarction. The study rationale is excellent the investigators are superbly qualified to carry out this important early phase trial. The authors should acknowledge the less distinct differentiation and function of regulatory T cells in humans versus mice. (see for example: Shevach EM. Biological functions of regulatory T cells. Adv Immunol 2011;112:137-76.) One issue of concern in the design is the safety monitoring proposal. A "blinded" trial management group (TMG) will apparently initially review adverse events. The chief investigator of the trial and other trial personnel in addition to an unblinded study statistician will comprise the TMG. To preserve the blind of the study and to assure the most rigorous conduct of the trial, shouldn't such analyses be performed by the "independent data monitoring committee." What is the composition of this latter committee? Is it fully independent from the investigative team? Shouldn't this independent group perform the primary safety analysis, and shouldn't an unblinded statistician not work closely with the investigators in adverse event analysis? I would ask the authors to clarify the relationship of the trial management group to the independent data safety monitoring board, as I view understanding this relationship is pivotal to the ability of readers to judge the rigor of the assessment of adverse effects and protection of human subjects. As therapeutic doses of IL-2 have been associated with profound capillary leak syndrome and IL-2 has long been known to lead to
---

	endothelial cell injury, vigilance for adverse effects is particularly important. (See for example: Aronson FR, Libby P, Brandon EP, Janicka MW, Mier JW. Interleukin-2 rapidly induces natural killer cell adhesion to human endothelial cells: A potential mechanism for endothelial injury. J Immunol 1988;141:158-163.)
--	---

REVIEWER	Nikolaos Frangogiannis Albert Einstein College of Medicine, USA
REVIEW RETURNED	20-Apr-2018

GENERAL COMMENTS	This is a very well-written protocol describing a double blind placebo-controlled trial examining the safety profile and tolerability of low dose IL-2 (aldesleukin) in patients with Ischemic Heart Disease and non-STEMI. An important goal of the study is to determine the dose of IL-2 that increases circulating Tregs by at least 75%. The rationale is well-described. Several minor concerns need to be addressed:  1. The inclusion criteria are ambiguous. Although the authors indicate that patients with IHD will be enrolled, in the table the only relevant inclusion criterion is evidence of “coronary artery disease”. First, CAD can be asymptomatic and does not necessarily cause ischemia. Will patients with objective evidence of ischemia (or ischemic symptoms) in the presence of CAD will be enrolled? Second, please clarify (at least in the table) the definition of significant coronary disease, if coronary arteriography is used. Third, in the text, the brief summary of the inclusion and exclusion criteria only includes exclusion criteria. Please revise to indicate the main inclusion criteria. 2. In the table listing the inclusion criteria, please briefly define diagnostic criteria for NSTEMI. 3. Outcome measures: Please provide specific information on the outcome measures. Which electrolytes will be assessed? Which tests constitute the bone profile? Which LFTs? Which clotting tests? Which vital observations? 4. Following MI, there may be dynamic changes in circulating Treg numbers. Thus, the timing of administration in relation to the acute event may be an important determinant of the effects of IL-2. Do the authors plan to start administration at a specific timepoint?
---

VERSION 1 – AUTHOR RESPONSE

Reply to reviewers:

Reviewer 1:

The authors would like to thank Prof. Libby for the careful and thorough reading of this manuscript and for the thoughtful comments and constructive suggestions, which help to improve the quality of this manuscript. Our response follows:

Reviewer: The authors should acknowledge the less distinct differentiation and function of regulatory T cells in humans versus mice.

Reply: We agree that human regulatory T cells have less distinct differentiation and function and have now commented accordingly in the background section.

Reviewer: A "blinded" trial management group (TMG) will apparently initially review adverse events. The chief investigator of the trial and other trial personnel in addition to an unblinded study statistician will comprise the TMG. To preserve the blind of the study and to assure the most rigorous conduct of the trial, shouldn't such analyses be performed by the "independent data monitoring committee."

Reply: The unblinded statistician presents analyses to the blinded TMG in an aggregated and not individualised format to preserve blinding for the other TMG members. This format is often used in early phase clinical trials where dose escalation is primarily based on safety from clinical events. The unblinded and independent data monitoring committee (DMC) will review all adverse events (AEs) and available data before progression to Part B (acute ACS patients). If there are any concerns or uncertainty raised by the TMG then they can refer to the DMC for further assessment. In addition, a DMC meeting is triggered at any stage when two patients within the same group experience any combination of a serious adverse reaction (SAR), an AE that is severe and at least possibly related to the trial drug, or any of the objective stopping criteria listed in the protocol. The DMC will then review all available unblinded data. We have amended the manuscript to provide further clarity and submitted the DMC charter as a supplementary document which outlines procedures and roles in more detail.

Reviewer: What is the composition of this latter committee? Is it fully independent from the investigative team?

Reply: The DMC will be comprised of a fully independent group of clinical researchers with suitable experience in experimental medicine and early phase clinical trials. These researchers are independent from the trial team and have not been involved in the setup or running of this clinical trial. We have amended the manuscript to provide further clarity and submitted the DMC charter as a supplementary document which outlines procedures and roles in more detail.

Reviewer: Shouldn't this independent group perform the primary safety analysis, and shouldn't an unblinded statistician not work closely with the investigators in adverse event analysis? I would ask the authors to clarify the relationship of the trial management group to the independent data safety monitoring board, as I view understanding this relationship is pivotal to the ability of readers to judge the rigor of the assessment of adverse effects and protection of human subjects.

Reply: To clarify, the TMG and the DMC are completely independent of each other. The Chief Investigator is present at DMC meetings only to explain to the DMC the course of the trial to date. All proceedings of the DMC are conducted privately and independent of the TMG. The unblinded statistician will work closely with the DMC to analyse all available data in an unblinded manner. The DMC will assess the safety of the trial between Part A and Part B, and also be convened if any 2 serious events occur in the same group of patients, and report these findings to the trial group to determine the course of the trial. A separate paragraph explaining this is now included in the manuscript, as well as further clarity elsewhere. We have also submitted the DMC charter as a supplementary document which outlines procedures, roles and relationships in more detail.

Reviewer: As therapeutic doses of IL-2 have been associated with profound capillary leak syndrome and IL-2 has long been known to lead to endothelial cell injury, vigilance for adverse effects is particularly important

Reply: We acknowledge that at much higher doses than proposed in this trial, capillary leak syndrome can occur. Accordingly, we have included this in our risk mitigation strategy which is now explained more extensively in the manuscript. This risk mitigation strategy will now be included in Supplementary Materials.

Reviewer 2:

The authors would like to thank Prof. Frangogiannis for the careful and thorough reading of this manuscript and for the thoughtful comments and constructive suggestions, which help to improve the quality of this manuscript. Our response follows in numerical order in reply to each of the reviewer's points.

1. The full details of the inclusion and exclusion criteria for Part A, and separately for Part B, have been elaborated on in the manuscript as requested. In reply to the specific question, patients in Part A will either have a previous diagnosis of myocardial infarction or, symptoms of angina and a coronary angiogram showing obstructive (stenosis >50%) coronary disease. Part B patients will be those currently admitted with an acute coronary syndrome. As the drug is currently contraindicated in this population, it was necessary to pick more stable patients in Part A without ongoing ischemia in order to provide proof that it was safe to progress into a more unwell population. We have clarified this in the Table 1 with the text taken as verbatim from the protocol.

2. The definition has been clarified to match the exact protocol wording: Current admission (on at least screening visit) with acute coronary syndrome (non-ST elevation myocardial infarction, i.e., NSTEMI, or unstable angina) with symptoms of myocardial ischaemia lasting 10 minutes or more, with the patient at rest or with minimal effort, plus either elevated levels of Troponin I on admission or dynamic changes in ECG (new ST-T changes) or T-wave inversion.

3. The outcome measures in the manuscript has been expanded upon to list the specific tests performed.

4. The authors agree with the reviewer's premise and our protocol inclusion criteria specifically state that dosing of the patient must start within 8 days of admission. This is based on pragmatism of consenting patients in the context of an admission with an acute coronary syndrome and potential transfer to the interventional hospital, and in keeping with the known management of such patients. However, we agree with the reviewer, and we will endeavour to maintain as much consistency between onset of dosing with the date of presentation with MI.

VERSION 2 – REVIEW

REVIEWER	Peter Libby Brigham and Womens Hospital and he Harvard Medical School. USA
REVIEW RETURNED	25-Jun-2018

GENERAL COMMENTS	The authors explanations and the revisions in the text have fully responded to my concerns and queries. The authors have embarked on important investigation with the aim of translating a large body of experimental and preliminary human data to individuals with acute and chronic coronary artery disease.
---

REVIEWER	Nikolaos Frangogiannis
-----------------	------------------------

	Albert Einstein College of Medicine, USA
REVIEW RETURNED	11-Jun-2018
GENERAL COMMENTS	The authors have addressed all my concerns.